# Modified FOLFIRINOX as a Second-Line Treatment for Patients with Gemcitabine-Failed Advanced Biliary Tract Cancer: A Prospective Multicenter Phase II Study

**DOI:** 10.3390/cancers14081950

**Published:** 2022-04-13

**Authors:** Yong-Pyo Lee, Sung Yong Oh, Kwang Min Kim, Se-Il Go, Jung Hoon Kim, Seok Jae Huh, Jung Hun Kang, Jun Ho Ji

**Affiliations:** 1Division of Hematology-Oncology, Department of Medicine, Samsung Changwon Hospital, Sungkyunkwan University School of Medicine, Changwon 51353, Korea; knee49@naver.com; 2Division of Hematology and Oncology, Department of Internal Medicine, Dong-A University Hospital, Dong-A University College of Medicine, Busan 49201, Korea; drosy@dau.ac.kr; 3Division of Gastroenterology, Department of Medicine, Samsung Changwon Hospital, Sungkyunkwan University School of Medicine, Changwon 51353, Korea; argonkim@gmail.com; 4Division of Hematology/Oncology, Department of Internal Medicine, Gyeongsang National University Changwon Hospital, Gyeongsang National University College of Medicine, Changwon 51472, Korea; gose1@hanmail.net; 5Division of Hematology/Oncology, Department of Internal Medicine, Gyeongsang National University Hospital, Jinju 52727, Korea; johnvankim@gmail.com; 6Department of Internal Medicine, Dong-A University College of Medicine, Busan 49201, Korea; doctorhsj@naver.com; 7Division of Hematology/Oncology, Department of Internal Medicine, College of Medicine Gyeongsang National University, Jinju 52727, Korea

**Keywords:** biliary tract cancer, second-line treatment, modified FOLFIRINOX

## Abstract

**Simple Summary:**

Biliary tract cancer is a malignant tumor of the biliary tract and gallbladder. Most patients are diagnosed at an advanced stage, and the basis of treatment is combination chemotherapy. However, the survival outcomes for biliary tract cancer, especially for patients who have failed frontline treatment, are poor. Accordingly, there have been various studies on effective subsequent treatments, and this study is one of those efforts. Through this study, we attempted to demonstrate the efficacy of enhanced chemotherapy in relapsed or refractory biliary tract cancer.

**Abstract:**

Background: After the publication of the ABC-02 trial, gemcitabine and cisplatin combination therapy (GP) became the standard first-line treatment for advanced biliary tract cancer (BTC). Despite GP therapy, most patients suffer from disease progression. The ABC-06 trial recommended FOLFOX as a second-line treatment, but its efficacy was modest. In this phase II study, we looked at the efficacy and safety of a second-line modified dose of FOLFIRINOX (mFOLFIRINOX) for patients who had failed first-line gemcitabine-based treatment. Methods: From January 2020 to January 2021, 34 patients with advanced BTC who failed first-line gemcitabine-based chemotherapy were enrolled. We evaluated the clinical efficacy and safety outcomes of mFOLFIRINOX. Results: With a median follow-up duration of 13.4 months, the median progression-free survival and overall survival was 2.8 months (95% confidence interval (CI): 1.6–4.0 months) and 6.2 months (95% CI: 5.0–7.4 months), respectively. The objective response rate was 14.7% with no complete response. The disease control rate was 61.7%, with a disease control duration of 4.2 months. Due to the rapid progression of the disease, approximately half of all patients received less than three cycles of treatment. The most common type of adverse event (AEs) was hematopoietic AEs. The incidence of non-hematopoietic AEs was relatively low. Conclusions: The efficacy of mFOLFIRINOX as a second-line treatment in advanced BTC patients after the failure of gemcitabine-based first-line treatment was replicated, albeit with slightly shorter survival results compared to previous studies. Long-term administration of mFOLFIRINOX with toxicity management might offer a survival benefit.

## 1. Introduction

Biliary tract cancers (BTCs) are heterogeneous malignancies, including intrahepatic cholangiocarcinoma (IHCCC), extrahepatic cholangiocarcinoma (EHCCC), and gallbladder (GB) cancer. Because BTC is frequently asymptomatic in the early stages, is difficult to access for biopsy, and the diagnostic imaging criteria are unclear, most patients are diagnosed at an advanced stage [1]. The role of systemic chemotherapy has been actively studied in advanced BTC because the effect of local treatments is meager [2,3,4]. After the publication of the ABC-02 trial [5], the combination of gemcitabine and cisplatin (GP) became the most widely used first-line treatment for advanced BTC. Despite GP therapy, however, most patients’ disease progression continues, while some patients maintain their general health and can tolerate subsequent treatments [6,7]. As a result, there is a need for salvage treatments for these patients. Second-line chemotherapy for BTC has mainly been performed with drugs such as 5-fluorouracil (5-FU) or gemcitabine, but their efficacy was based on retrospective studies with small sample sizes, and even these results were not satisfactory [8,9]. According to the findings of the ABC-06 trial, FOLFOX therapy, which consists of 5-FU, leucovorin, and oxaliplatin, is now considered a standard second-line treatment for relapsed BTC [10]. However, as the survival benefit of FOLFOX is relatively modest, effective treatment options are still needed. Recently, many studies for immunotherapy and molecular targeted therapy have been introduced. Nonetheless, the role of immunotherapy in BTC is limited only to some patients with mismatch repair and microsatellite instability [11,12,13,14], and the prevalence of novel molecular targets, such as isocitrate dehydrogenase 1 (IDH1), fibroblast growth factor receptor 2 (FGFR2), human epidermal growth factor receptor 2 (HER2), and v-Raf murine sarcoma viral oncogene homolog B (BRAF), is approximately less than 20%, and heterogeneous by type of BTC [15,16,17,18,19]. As a result, many studies on the role of cytotoxic chemotherapy in BTC salvage treatment are still ongoing. FOLFIRINOX is an intensified cytotoxic chemotherapy used to treat pancreatic cancer that is adjacent to the biliary tract and has histological and molecular characteristics similar to BTC [20]. Accordingly, clinical studies using FOLFIRINOX as a first-line or salvage-line treatment were conducted in BTC [21,22,23,24,25]. In this study, we conducted a phase II study to assess the efficacy and safety of a modified dose of FOLFIRINOX therapy (mFOLFIRINOX) as a second-line treatment for Korean patients with BTC who had failed gemcitabine-based therapy.

## 2. Methods

### 2.1. Study Design and Treatment

This was a single-arm, multicenter, open-label prospective phase II study. Patients were treated with mFOLFIRINOX (oxaliplatin 65 mg/m^2^, irinotecan 135 mg/m^2^, and leucovorin 400 mg/m^2^, followed by continuous infusion of 1000 mg/m^2^/day of 5-FU for two consecutive days) every two weeks. The study treatment was intended to be administered for a minimum of three cycles and a maximum of twelve cycles, but it could be extended at the discretion of the investigator. In the case of disease progression, the occurrence of unacceptable toxicities, or withdrawal of consent, the study treatment was discontinued. In the case of grade 3/4 toxicities, treatment was postponed until the toxicities improved to grade 2 or lower, and when treatment was resumed, a 25% dose reduction was performed. Additional dose reduction was permitted if grade 3/4 toxicity persisted after the first dose adjustment. The study was terminated in patients who had delayed treatment for more than three weeks or who were considered to be unsuitable for further treatment. The granulocyte colony-stimulating factor (G-CSF) could be used in patients with febrile neutropenia or severe infection, but its prophylactic use was not permitted. Every three cycles of treatment, a response evaluation using computed tomography or magnetic resonance imaging was carried out following the Response Evaluation Criteria in Solid Tumors (RECIST) [26]. Toxicities were assessed at each treatment cycle according to the National Cancer Institute Common Terminology Criteria for Adverse Events, version 5.0 [27].

### 2.2. Eligibility

The main eligibility criteria were histologically confirmed adenocarcinoma of the biliary tract, which relapsed after the failure of gemcitabine-based chemotherapy. Other eligibility criteria were the following: between the ages of 19 and 75 years, Eastern Cooperative Oncology Group (ECOG) performance status 2 or lower, at least one measurable lesion based on RECIST criteria, and adequate organ function (absolute neutrophil count ≥1.5 × 10^9^/L, platelets ≥100 × 10^9^/L, bilirubin ≤1.5 × upper limit of normal (ULN), aspartate transaminase and/or alanine transaminase ≤3 × ULN, creatinine ≤1.5 mg/dL or creatinine clearance ≥50 mL/min, and ejection fraction ≥50% measured by echocardiography). The following were key exclusion criteria: BTC eligible for curative surgical treatment or radiation therapy, receiving chemotherapy or radiation therapy within two weeks or three weeks of registration, uncontrolled lesions in the central nervous system, severe cardiopulmonary comorbidity and/or active infection, pregnant or lactating women, and hypersensitivity reaction to the drugs. This study was approved by the institutional review board of each institution, and the study was registered with the Clinical Research Information Service of Korea (CRIS, KCT0004433). All patients signed written informed consent.

### 2.3. Quality of Life Measurement

The quality of life (QOL) was assessed with the use of a cancer-specific, 30-item score questionnaire, QLQ-C30-Version 3.0 [28], and a 21-item questionnaire, QLQ-BIL21 [29], which was developed by the European Organisation for Research and Treatment of Cancer (EORTC). The EORTC QOL group procedures were used to transform and analyze the questionnaire items and scales. As a baseline, patients completed the questionnaire one week before receiving their first dose of treatment, and then one week before beginning subsequent treatment every three cycles.

### 2.4. Statistical Analysis

The primary objective of this study was to evaluate the median progression-free survival (PFS) of mFOLFIRINOX. Secondary goals included estimating the median overall survival (OS) and assessing the objective response rate (ORR), disease control rate (DCR), and treatment safety profiles. PFS time was estimated as the time from enrollment in the study to the date of disease progression or death related to any cause. The OS time was calculated as the time from study enrollment to the date of death or the last follow-up date. ORR was defined as the proportion of patients who achieved a complete response (CR) or partial response (PR) as their best response to treatment, and DCR was defined as the proportion of patients with CR, PR, and stable disease. According to Simon, the study sample size was determined as a phase II [30], optimal two-stage study with PFS at six months as the primary endpoint. Published historical outcomes of PFS at 6 months for second-line treatment of BTC ranged from 17% to 40% [16,31,32]. Thus, this phase II study was designed to detect an increase in 6 months PFS from 10% to 30%. The alpha error was set to 0.05 and the power was set to 0.8. Then, we calculated the number of needed subjects under a hypothesis of interest in which mFOLFIRINOX PFS at six months was >30% (H1 = 30%) and a null hypothesis in which it reached a PFS at six months <10% (H0 = 10%). At least one responder among the first ten subjects was needed, and at the second stage, a responder was needed for 29 subjects. We decided to include at least 33 patients to account for nonassessable subjects. Proportions were determined using standard descriptive statistics, and medians for categorical variables were assessed using the Mann–Whitney U test, paired *t*-test, or Fisher’s exact test. The Kaplan–Meier method was used to evaluate PFS and OS. All data were analyzed using the Statistical Package for Social Sciences software, version 24.0 (IBM Corp., Armonk, NY, USA).

## 3. Results

### 3.1. Patient Characteristics

Between January 2020 and January 2021, we enrolled a total of 34 patients with BTC who failed gemcitabine-based first-line chemotherapy. Baseline characteristics at the time of registration are shown in Table 1. The median age across all patients was 65 years (range, 40–78). There were 13 patients with IHCCC, 13 with EHCCC, and 8 with GB cancers among the 34 patients. All patients received GP therapy as first-line chemotherapy. The median number of GP cycles was five (range, 1–33) and the median PFS was 4.3 months. In each group, seven patients received GP therapy for more than six months or less than three months, while half of the patients received GP therapy for more than three months but less than six months.

### 3.2. Clinical Outcomes of mFOLFIRINOX

With a median follow-up duration of 13.4 months, the median PFS and OS were 2.8 months (95% confidence interval (CI): 1.6–4.0 months) and 6.2 months (95% CI: 5.0–7.4 months), respectively (Figure 1A,B). The median number of treatment cycles was four (range, 1–22). The response to treatment was assessed in all 34 patients, including three who died before the first response evaluation (two patients died of infection, and one withdrew from the study due to hypersensitivity to oxaliplatin). Five patients achieved PR and sixteen patients obtained SD, while thirteen patients did not respond to treatment. The ORR and DCR were 14.7% and 61.7%, respectively. The median duration of disease control was 4.2 months. The largest changes in tumor volume from baseline are shown in Figure 2. Better clinical outcomes were observed in patients who achieved objective responses or at least obtained disease control (Appendix A). Additionally, when patients were divided into three groups according to the number of mFOLFIRINOX administrations (that is, 3 or less, 3 to 12 or less, and more than 12), the patients who received a higher number of treatments had better survival outcomes (Appendix A). There was no statistically significant difference in survival according to the type of BTC or the duration of first-line GP therapy (Appendix A).

### 3.3. Safety Outcomes

The most common type of toxicity was hematopoietic adverse events (AEs). In particular, approximately one-third of patients experienced grade 3/4 neutropenia, of which more than 50% of patients had dose reduction in treatment (Table 2). In our study, 44% (*n* = 15) of patients received dose reductions, of which 40% (*n* = 6) experienced two dose reductions (50% reduction in total dose). The incidence of grade 3/4 non-hematopoietic AEs was low in comparison, resulting in an approximately 17% dose reduction in total patients. At the end of the follow-up, all patients had discontinued the study treatment, and most of the cases of mFOLFIRINOX discontinuation were due to disease progression (*n* = 25, 80.6%). During the study period, three deaths were reported, but all were caused by infections, and no deaths were caused by treatment.

### 3.4. Quality of Life

After measuring the baseline QOL before initiation of study treatment, the QOL scores at the time of maximal disease control were obtained from the patients who achieved disease control. Following that, QOL data were collected at the end of treatment to compare the changes in QOL scores over time. A total of 21 patients (61.7%) were included in the QOL analysis, with data collected at three time points (baseline, at the time of maximal disease control, and at the end of treatment) for 16 patients, excluding five patients who were unable to answer the questions due to the study’s unavoidable discontinuation. The number of patients who completed QOL measurements continued to decrease as the study progressed; however, subjective QOL improved with disease control in most patients (Table 3). Unfortunately, with disease progression, overall deterioration in QOL of the patients was observed.

## 4. Discussion

This is a prospective phase II study that evaluated the efficacy and safety of mFOLFIRINOX as a second-line treatment in Korean advanced BTC patients who failed gemcitabine-based treatment. The dose of mFOLFIRINOX used in our study was lower than that of the previous studies, and even that dose reduction occurred in 40% of total patients, but the ORR and DCR were not significantly different from previous studies (Appendix A) [23,24]. Although survival outcomes were somewhat disappointing, it should be noted that half of the patients received less than three cycles of study treatment because disease progression was so rapid, which resulted in shorter PFS and OS. Survival analysis according to the number of mFOLFIRINOX administrations showed that patients who received more cycles of treatment had longer PFS and OS (Appendix A). The incidence of dose reduction also increased with the increase in the number of treatments (*p*-value < 0.001), but it did not affect the survival outcomes in patients who received more than three cycles of treatment.

In our study, the most common type of AEs were hematopoietic toxicities, including neutropenia and anemia (Table 2). Grade 3/4 hematopoietic toxicities were frequently detected in the absence of a bolus dosage of 5-FU that was followed by myelosuppression. Hematopoietic toxicities resulted in dose reduction in the treatment in 20% of all patients, and also prolongation of treatment intervals. As a result, this needs to be controlled to some extent through adjuvant treatments, such as G-CSF or erythropoietin [33]. In a study on the effect of prophylactic pegylated G-CSF in front-line FOLFIRINOX for metastatic pancreatic cancer, researchers found that using pegylated G-CSF increased the cumulative relative dose intensity (cRDI), reduced neutropenia, and even improved survival rates [34]. Furthermore, considering the results of a study in which cRDI of FOLFIRINOX was one of the independent prognostic factors for survival [35], the prophylactic use of G-CSF was expected to contribute to stable administration of FOLFIRINOX.

Recently, with advances in the understanding of the molecular landscape of BTC, various studies have been conducted on precisional approaches, using targeted agents and immune checkpoint inhibitors. In particular, FGFR inhibitor [17] and IDH1 inhibitor [16] are in the spotlight, and a slew of clinical trials using a variety of actionable target molecules are currently underway. Despite the development of various novel agents for BTC, realistic problems such as a somewhat unsatisfying expression rate of molecular targets and cost burdens for novel agents are difficult obstacles. As a result, further research into the intensification of cytotoxic chemotherapy is required. FOLFIRINOX is a widely studied chemotherapy regimen for BTC, which stems from its usefulness in the treatment of pancreatic cancer [36,37,38]. Unlike pancreatic cancer, however, FOLFIRINOX had little efficacy in BTC, which was in line with a recently published phase II study comparing the efficacy of GP and mFOLFIRINOX [25]. Similarly, in this study, intensified chemotherapy elicited some tumor responses, but not survival benefits. Nevertheless, the results of the NIFTY trial [39] demonstrated that the combination of oxaliplatin and irinotecan with 5-FU still may be an attractive option as a salvage therapy for BTC. Thus, in order for FOLFIRINOX to be one of the most appealing options for advanced BTC treatment, an adequate therapeutic dose must be established, and patients who are predicted to benefit from treatment must be identified. Compared to the previous original regimen of FOLFIRINOX, the survival outcomes and response rate of mFOLFIRINOX were comparable and presented fewer AEs in pancreatic cancer treatment, leading to the use of a dose-attenuated FOLFIRINOX regimen in practice as well as clinical studies [40]. However, because FOLFIRINOX dosage modification is not standardized, several variants of the dose-adjusted treatment are used. As this is also the case with BTC, finding the optimal dose of FOLFIRINOX in BTC treatment should be studied in the future. Furthermore, the identification of prognostic and predictive factors aimed at administering FOLFIRINOX to appropriate patients is required. Several studies have revealed that levels of carbohydrate antigen 19-9 or the maximal standardized uptake value of 18F-fluorodeoxyglucose positron emission tomography are predictive variables for FOLFIRINOX therapy in pancreatic cancer treatment [41,42]. As a result, future studies on prognostic and predictive factors for FOLFIRINOX therapy in BTC are needed, and more attention is needed in selecting patients who will receive the maximum benefit from treatment.

There are several limitations in our study: (1) the sample size was not large enough to draw some exceptional conclusions, (2) we could not specify the characteristics of patients who experienced rapid disease progression, and (3) in the absence of molecular study, we were unable to identify predictive markers for a good response to mFOLFIRINOX therapy.

## 5. Conclusions

The efficacy of mFOLFIRINOX in advanced BTC as a second-line treatment after the failure of gemcitabine-based treatment was reproduced. In future research, it will be necessary to improve cRDI with effective drug delivery in addition to managing and preventing AEs following mFOLFIRINOX. Moreover, to select optimal patients for mFOLFIRINOX, it is important to find biological markers that can predict treatment responses.

## Figures and Tables

**Figure 1 cancers-14-01950-f001:**
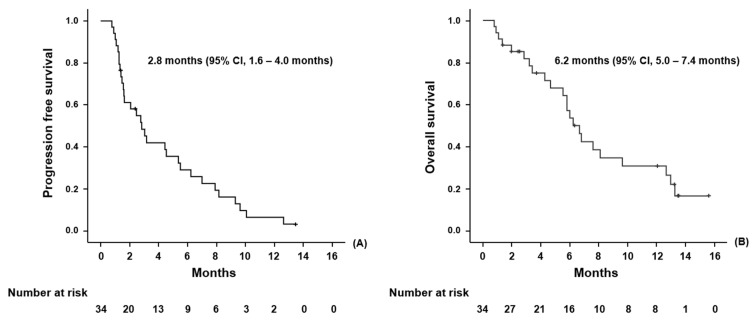
Kaplan–Meier curves for progression-free survival (**A**) and overall survival (**B**).

**Figure 2 cancers-14-01950-f002:**
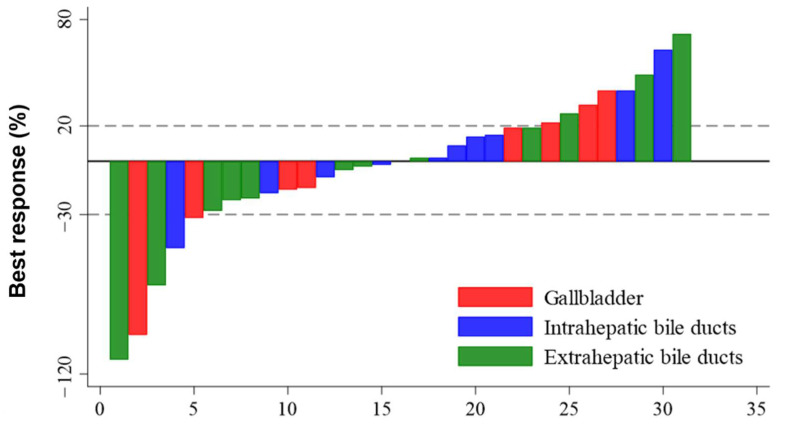
Waterfall plot of modified FOLFIRINOX according to the best response.

**Table 1 cancers-14-01950-t001:** Baseline demographics and disease characteristics at the time of enrollment (*n* = 34).

Characteristics		*n* (%)
Age, year		
Median, range		65 (40–78)
Sex		
Male		28 (82.3%)
Female		6 (17.6%)
ECOG performance status		
0		8 (23.5%)
1		26 (76.4%)
2		0 (0.0%)
Primary tumor site		
Intrahepatic bile duct		13 (38.2%)
Extrahepatic bile duct		13 (38.2%)
Gallbladder		8 (23.5%)
Differentiation		
Well-differentiated		6 (17.6%)
Moderate-differentiated		11 (32.3%)
Poorly differentiated		6 (17.6%)
Unknown		11 (32.3%)
Baseline tumor markers according to the primary site		
Intrahepatic bile duct	CEA (ng/mL), mean ± SD	83.2 ± 254.0
	CA 19-9 (U/mL), mean ± SD	2041.0 ± 3541.8
Extrahepatic bile duct	CEA (ng/mL), mean ± SD	92.1 ± 270.1
	CA 19-9 (U/mL), mean ± SD	2183.7 ± 3713.9
Gallbladder	CEA (ng/mL), mean ± SD	50.8 ± 151.5
	CA 19-9 (U/mL), mean ± SD	1686.0 ± 3269.8
Type of first-line chemotherapy		
Gemcitabine plus Cisplatin		34 (100.0%)
Median number of first-line chemotherapy, range		5 (1–33)
Duration of first-line chemotherapy		
<3 months		7 (20.5%)
3 months ≤ 6 months		18 (52.9%)
6 months ≤		7 (20.5%)
Unknown		2 (5.8%)

ECOG, Eastern Cooperative Oncology Group; CEA, Carcino Embryonic Antigen; CA 19-9, Carbohydrate antigen 19-9.

**Table 2 cancers-14-01950-t002:** Adverse events of mFLOFIRINOX treatment.

	Number of Patients (%)			
Hematopoietic Adverse Events	All grades	Grade 1	Grade 2	Grade 3–4
Neutropenia	23 (67.6)	8 (23.5)	4 (11.7)	11 (32.3)
Anemia	28 (82.3)	8 (23.5)	11 (32.3)	9 (26.4)
Thrombocytopenia	17 (50.0)	11 (32.3)	1 (2.9)	5 (14.7)
Non-hematopoietic adverse events	All grades	Grade 1	Grade 2	Grade 3–4
Febrile neutropenia	3 (8.8)	0 (0.0)	0 (0.0)	3 (8.8)
Liver enzyme elevation	8 (23.5)	5 (14.7)	3 (8.8)	0 (0.0)
Azotemia	4 (11.7)	4 (11.7)	0 (0.0)	0 (0.0)
Emesis	11 (32.3)	8 (23.5)	3 (8.8)	0 (0.0)
Anorexia	6 (17.6)	3 (8.8)	2 (5.8)	1 (2.9)
Diarrhea	8 (23.5)	5 (14.7)	2 (5.8)	1 (2.9)
Stomatitis	6 (17.6)	3 (8.8)	2 (5.8)	1 (2.9)
Sensory neuropathy	8 (23.5)	4 (11.7)	2 (5.8)	2 (5.8)

mFOLFIRINOX, modified FOLFIRINOX.

**Table 3 cancers-14-01950-t003:** Changes in the quality of life of the patients, presented as absolute values.

Items	Baseline	Best Response		Progression	
	Mean ± SD	Mean ± SD	*p*	Mean ± SD	*p*
Physical function	9.52 ± 3.78	8.24 ± 3.63	0.008	11.19 ± 3.75	0.012
Role function	4.14 ± 1.90	3.71 ± 1.82	0.117	4.81 ± 2.04	0.035
Emotional function	7.14 ± 1.60	6.29 ± 1.62	0.015	7.88 ± 2.85	0.260
Cognitive function	2.95 ± 0.92	2.62 ± 0.80	0.095	3.56 ± 1.31	0.053
Social function	3.24 ± 1.09	3.24 ± 1.09	0.867	4.13 ± 1.67	0.016
Symptom scale	21.95 ± 6.41	19.67 ± 4.68	0.031	24.06 ± 6.29	0.013
Financial scale	1.67 ± 0.58	1.48 ± 0.60	0.157	1.75 ± 0.58	0.480
Global health status	10.67 ± 8.40	8.95 ± 1.40	0.470	9.06 ± 2.05	0.521
Biliary symptom	36.00 ± 6.28	33.00 ± 5.04	0.134	38.94 ± 6.80	0.046

SD, standard deviation.

## Data Availability

The data that support the findings of this study are available on request from the corresponding author.

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
