# Peer review of "Modified FOLFIRINOX as a Second-Line Treatment for Patients with Gemcitabine-Failed Advanced Biliary Tract Cancer: A Prospective Multicenter Phase II Study"

_cancers, 2022, doi:10.3390/cancers14081950_

Round 1
Reviewer 1 Report
The authors analyse mFOLFIRINOX as a second-line treatment for patients with gemcitabine-failed advanced biliary tract cancer (BTC) in a prospective multicentre phase II trial. This is an interesting analysis of a clinical relevant topic. Obviously, there are data for mFOLFIRINOX as second-line or salvage therapy in BTC. Thus, there is lack of novelty. Nevertheless, the data are valid and important.
The primary endpoint was median progression-free survival. The power calculation is not clear and should be better explained.
How often was imaging and/or clinical staging performed? The median duration of disease control was 4.2 months and PFS 2.8 months. If imaging was performed for example every 3 months, it is difficult to estimate those rates. Please comment.
“The patient group that received more than three cycles of treatment had a significant survival benefit compared to the group that did not”. One could also argue that patients that survived longer could receive more cycles, i.e. there is no proof of a benefit.
Author Response
The authors analyse mFOLFIRINOX as a second-line treatment for patients with gemcitabine-failed advanced biliary tract cancer (BTC) in a prospective multicentre phase II trial. This is an interesting analysis of a clinical relevant topic. Obviously, there are data for mFOLFIRINOX as second-line or salvage therapy in BTC. Thus, there is lack of novelty. Nevertheless, the data are valid and important.
þ We appreciate the time and effort you have dedicated to providing insightful feedback on ways to strengthen our paper. To facilitate your review of our revisions, the following is a point-by point response to the questions and comments.
1) The primary endpoint was median progression-free survival. The power calculation is not clear and should be better explained.
þ Thanks for providing this insight. In general, as you mentioned, overall response rate rather than PFS is widely used to set the sample size for phase II study. However, in BTC, it is hard to evaluate the treatment response clearly due to the ambiguous boundary of primary tumor with its invasion to surrounding tissue [1]. And even BTC patients who do not show an objective response to chemotherapy can sometimes survive long time, so we focused on PFS. The PFS of chemotherapy for second-line treatment of BTC is generally quite short, and there is considerable variation among studies. Six month PFS showed more consistent results than PFS, and was important in the prognosis of BTC [2-4] (44% to 61% in first-line treatment [5-7] and 17% to 40% in second-line treatment [8-10], respectively). Thus, we set the 6 month PFS of mFOLFIRINOX to 30% and PFS as a primary endpoint rather than response rate. In manuscript, we have clarified it.
þ Lines 141-143, Page 3, after revision
Published historical outcomes of PFS at 6 months for second-line treatment of BTC ranged from 17% to 40% [16, 30, 31]. Thus, this phase II study was designed to detect an increase in 6 months PFS from 10% to 30%.
2) How often was imaging and/or clinical staging performed? The median duration of disease control was 4.2 months and PFS 2.8 months. If imaging was performed for example every 3 months, it is difficult to estimate those rates. Please comment.
þ Thank you for your comments. The study treatment was repeated every two weeks and we evaluated the disease status every three cycles. Therefore, imaging test including CT or MRI were performed every 1.5 months.
3) “The patient group that received more than three cycles of treatment had a significant survival benefit compared to the group that did not”. One could also argue that patients that survived longer could receive more cycles, i.e. there is no proof of a benefit.
þ We agree with your comment. In our study, patients who received less than three cycles of mFOLFIRINOX, so-called under-treated patients, mostly experienced rapid disease progression despite of treatment. However, patients with at least no disease progression with mFOLFIRINOX treatment achieved their disease control (at least SD/PR/CR) within a median of 1.3 months (95% CI, 1.1–1.5 months), and tended to maintain their disease status (median disease control 4.2 months (95% CI, 2.9–5.5 months). Thus, we believe that it is conceivable that they could be considered patients who benefited from mFOLFIRINOX treatment.

Reviewer 2 Report
This is a manuscript reporting results of a phase II study using FOLFIRINOX for patients with refractory cholangiocarcinoma. The results are not very encouraging and there is currently no standard therapy for these patients. The study was carried out in a standard manner and the manusript is well written.
- What is the main question addressed by the research?
The manuscript aims to test, in a single-arm study, the efficacy and safety of a combination chemotherapy regimen for biliary cancer.
2. Do you consider the topic original or relevant in the field, and if so, why?
The problem of second-line therapy for metastatic biliary cancer is unresolved and thus testing of new regimens is needed.
3. What does it add to the subject area compared with other published material?
There is paucity of data on the efficacy of second-line regimen and this paper provides OS and PFS data for FOLFIXIRI, on which our current knowledge is inadequate.
4. What specific improvements could the authors consider regarding the methodology?
This is a standard phase II trial and it was well done.
5. Are the conclusions consistent with the evidence and arguments presented and do they address the main question posed?
The conclusions are consistent with the evidence and they are well presented.
6. Are the references appropriate?
Yes.
7. Please include any additional comments on the tables and figures. Tables and figures adequately illustrate the findings and support the conclusions.
Author Response
Reviewer #2:
This is a manuscript reporting results of a phase II study using FOLFIRINOX for patients with refractory cholangiocarcinoma. The results are not very encouraging and there is currently no standard therapy for these patients. The study was carried out in a standard manner and the manusript is well written.
þ We appreciate the time and effort you have dedicated to providing insightful feedback on ways to strengthen our paper. To facilitate your review of our revisions, the following is a point-by point response to the questions and comments.
- What is the main question addressed by the research?
The manuscript aims to test, in a single-arm study, the efficacy and safety of modified FOLFIRINOX regimen as a salvage treatment for biliary cancer.
- Do you consider the topic original or relevant in the field, and if so, why?
The problem of second-line therapy for metastatic biliary cancer is unresolved and thus testing of new regimens is needed. And this is a clinical study on a modified dose of FOLFIRINOX, which is highly toxic, for patients with biliary tract cancer who have failed first line treatment, which is vulnerable to the toxicity of chemotherapy
- What does it add to the subject area compared with other published material?
There is paucity of data on the efficacy of second-line regimen and this paper provides OS and PFS data for modified FOLFIRINOX, on which our current knowledge is inadequate.
þ In our study, compared with the previously published FOLFIRINOX study [1], dose of FOLFIRINOX was significantly reduced, but serious toxicities were observed. Unlike other races, Asians may need more dose reduction when considering salvage chemotherapy for BTC.
- What specific improvements could the authors consider regarding the methodology?
Our study is a prospective single arm phase II study. If there was a double blind control group, more meaningful results could have been produced. However, our result for reduced dose of FOLFIRINOX from refractory cholangiocarcinoma will provide clinicians with a various treatment options in the future.
- Are the conclusions consistent with the evidence and arguments presented and do they address the main question posed?
The conclusion of our study is that modified FOLFIRINOX can be considered for refractory cholangiocarcinoma. This conclusion is drawn from the results of our phase II study.
- Are the references appropriate?
Yes.
- Please include any additional comments on the tables and figures.
Tables and figures adequately illustrate the findings and support the conclusions.
- Belkouz A, de Vos-Geelen J, Mathôt RAA, Eskens FALM, van Gulik TM, van Oijen MGH, Punt CJA, Wilmink JW, Klümpen H-J. Efficacy and safety of FOLFIRINOX as salvage treatment in advanced biliary tract cancer: an open-label, single arm, phase 2 trial. British Journal of Cancer. 2020;122(5):634-9.

Reviewer 3 Report
The authors recognised that ABC-06 FOLFOX has limited clinical benefit and set out to improve the outcomes with the intensification of chemotherapy with FOLFIRINOX but unfortunately without a standard arm for comparison. Overall well written, but some word choices could be improved. For instance, methods about Simon two-stage design wording is not in the usual wording style that is more commonly seen. For another example, line 208 – survival outcome of your study did not fair worse than the two retrospective studies. The way you have written can be ambiguous to mean that your study survival was worse when I think your actual intention is to mean that the survival outcomes are improved but not to any significant extent clinically.
I think the discussion needs a fair bit of revision. It is currently focused on why it is reasonable to reference pancreatic cancer and the need to optimise the dose of FOLFIRINOX to get a better outcome. However, the discussion should focus on discussing your results specifically. It is better to discuss how your results sit with the other two retrospective studies with reference to the clinical practice implications. Optimal dose-finding may be relevant but needs to balance that other studies done with higher doses and FOLFIRINOX in the first-line setting have not yielded better results. Therefore, I am not sure you will get practice-changing results however you optimise. The authors’ second line FOLFIRINOX yielded more toxicity than FOLFOX as per ABC06 without getting better survival should be acknowledged. Combined with PRODIGE 38 results, I am not sure further clinical pursuit of FOLFIRINOX makes sense. It would be good at this point to comment why the author chose a dosing of mFOLFIRINOX at lower than published doses.
Another suggestion is to have patient numbers under your Kaplan Mier curves at the set time intervals and include the supplemental figure you have referred to in lines 208-210 to be in the main text.
QoL measurement of which 5 patients were excluded because of “unavoidable discontinuation, " yet the study methods did not state the study was discontinued/ended prematurely. QoL statement in line 199 suggests that QoL improved with disease control. Whilst this is sound logically but the evidence portrayed in the article is unable to support that conclusion. If anything, the best response time point, emotional and physical functions have statistically significantly deteriorated. This is reasonable from a clinical perspective given FOLFIRINOX, but the data does not support the authors' statement.
Line 66/67 – disagree with authors’ statement that actionable mutation percentage is low because the like of FGFR, IDH and BRAF is a reasonable number of sections with good responses. The review they have based their statement on shows a sizeable percentage. Could authors please clarify their position and how it is “low”? Response to immunotherapy has been published in the firstline setting, and more up-to-date references should be included.
Line 72- reference 14-17 needs to be updated. PRODIGE 38 has been published in whole entirety.
Please do statistical calculation based on the entire population (i.e. 34 patients ) and not exclude the three who died before the first disease response assessment.
Author Response
Reviewer #3:
The authors recognised that ABC-06 FOLFOX has limited clinical benefit and set out to improve the outcomes with the intensification of chemotherapy with FOLFIRINOX but unfortunately without a standard arm for comparison. Overall well written, but some word choices could be improved. For instance, methods about Simon two-stage design wording is not in the usual wording style that is more commonly seen. For another example, line 208 – survival outcome of your study did not fair worse than the two retrospective studies. The way you have written can be ambiguous to mean that your study survival was worse when I think your actual intention is to mean that the survival outcomes are improved but not to any significant extent clinically.
þ Thank you very much for your precious review of our manuscript. All the comments were very helpful for correction of this report, and we did our best to accommodate your detailed advice as much as possible. In accordance with the respectful comments, we have revised our manuscript.
1) I think the discussion needs a fair bit of revision. It is currently focused on why it is reasonable to reference pancreatic cancer and the need to optimise the dose of FOLFIRINOX to get a better outcome. However, the discussion should focus on discussing your results specifically. It is better to discuss how your results sit with the other two retrospective studies with reference to the clinical practice implications. Optimal dose-finding may be relevant but needs to balance that other studies done with higher doses and FOLFIRINOX in the first-line setting have not yielded better results. Therefore, I am not sure you will get practice-changing results however you optimise. The authors’ second line FOLFIRINOX yielded more toxicity than FOLFOX as per ABC06 without getting better survival should be acknowledged. Combined with PRODIGE 38 results, I am not sure further clinical pursuit of FOLFIRINOX makes sense. It would be good at this point to comment why the author chose a dosing of mFOLFIRINOX at lower than published doses.
þ We appreciate your in-depth review, and we fully agree with your point. In our study, the dose of mFOLFIRINOX was determined by referring to the dose of mFOLFIRINOX used in pancreatic cancer in our previous study [1]. In the safety profile of most prospective studies on FOLFIRINOX in BTC, grade 3/4 toxicities are observed in more than 70% of subjects. Therefore, the dosage of mFOLFIRINOX in our study had been reduced from the dose used in previous studies, and even approximately 40% of patients underwent further dose reductions. This is probably because, in second-line treatment, patients are more vulnerable to the toxicities of combination chemotherapy. Thus, we thought that reducing the dose of mFOLFIRINOX with less toxicities would allow patients to be exposed to the study treatment for a longer period of time, thereby avoiding premature discontinuation of treatment. We agree that we have not demonstrated the advantages of adding Irinotecan to standard FOLFOX regimen with prolonged survivals. However, although we have not been able to discriminate, we believe that there will be patient groups who may benefit from irinotecan administration, and in some cases, we would like to regard the possibility that irinotecan-containing regimen may provide a bit of benefits. We will organize the doses of treatment and survival results used in previous studies and this study in a supplementary table, and add comments in the discussion session.
þ Lines 219-227, Page 7, after revision
: The dose of mFOLFIRINOX used in our study was lower than that of the previous studies and even that dose reduction occurred in 40% of total patients, but the ORR and DCR were not significantly different from previous studies (Table S1) [23, 24]. Although the survival outcomes were rather disappointing, with shorter PFS and OS. Although survival out-comes were somewhat disappointing with shorter PFS and OS, it should be considered that half of our patients presented disease progression so rapid that they did not receive less than three cycles of study treatment. The survival analysis according to the number of mFOLFIRINOX administrations showed that patients who received more cycles of treatment had longer PFS and OS (Figure S2A).
þ Lines 254-255, Page 8, after revision
: Similarly, in this study, intensified chemotherapy elicited some tumor responses, but not survival benefits.
þ Lines 276-277, Page 8, after revision
: 3) in the absence of molecular study, we were unable to identify predictive markers of good response to mFOLFIRINOX therapy.
2) Another suggestion is to have patient numbers under your Kaplan Mier curves at the set time intervals and include the supplemental figure you have referred to in lines 208-210 to be in the main text.
þ Thank you for your suggestion. According to your comments, we have provided the number at risk on survival curves and added supplementary figures showing survival curves according to the number of mFOLFIRINOX administrations. We thought that these figures could be replaced by a supplementary figure 2. Please check the figures.
3) QoL measurement of which 5 patients were excluded because of “unavoidable discontinuation, " yet the study methods did not state the study was discontinued/ended prematurely.
þ We are sorry that our phrases confused reviewers. The five patients mentioned were not included in the QOL assessment due to one death, one surgery for remnant disease after mFOLFIRINOX treatment, one unacceptable hematologic toxicity, and two withdrawal of consent, and we acknowledged discontinuation of study treatment for these reasons (Line 92-93, Page 2).
4) QoL statement in line 199 suggests that QoL improved with disease control. Whilst this is sound logically but the evidence portrayed in the article is unable to support that conclusion. If anything, the best response time point, emotional and physical functions have statistically significantly deteriorated. This is reasonable from a clinical perspective given FOLFIRINOX, but the data does not support the authors' statement.
þ We are sorry for the confusion caused by our insufficient elucidation. The values ​​shown in Table 3 were displayed as absolute values that was written by patients for each item, ​​without converting according to QLQ-30 or QLQ-BIL21 scoring scale. Therefore, changes in physical function, emotional function, and symptom scale when the best response was obtained can be interpreted as improved. To avoid confusion, it is stated that the values ​​in Table 3 are absolute values ​​assessed by patients.
þ Lines 214, Page 7, after revision
: Table 3. Changes in the quality of life of the patients, presented as absolute values.
5) Line 66/67 – disagree with authors’ statement that actionable mutation percentage is low because the like of FGFR, IDH and BRAF is a reasonable number of sections with good responses. The review they have based their statement on shows a sizeable percentage. Could authors please clarify their position and how it is “low”? Response to immunotherapy has been published in the firstline setting, and more up-to-date references should be included.
þ We acknowledge that the phrase "quite low" in the text is not sufficient. According to your comments, we added phrases for each mutation and their expression rate in the introduction session with references. Additionally, we updated references on the results of immunotherapy as a first-line treatment.
þ Lines 71-77, Page 2, after revision
: Nonetheless, the role of immunotherapy in BTC is limited only to some patients with mismatch repair and microsatellite instability [11], and the prevalence of novel molecular targets, such as isocitrate dehydrogenase 1 (IDH1), fibroblast growth factor receptor 2 (FGFR2), human epidermal growth factor receptor 2 (HER2) and v-Raf murine sarcoma viral onco-gene homolog B (BRAF) is approximately less than 20%, and heterogeneous by type of BTC [12].
þ Lines 248, Page 8, after revision
: somewhat unsatisfying expression rate of molecular targets
6) Line 72- reference 14-17 needs to be updated. PRODIGE 38 has been published in whole entirety.
þ Thank you for your comments. We updated reference article.
7) Please do statistical calculation based on the entire population (i.e. 34 patients ) and not exclude the three who died before the first disease response assessment.
þ According to your comments, we re-performed the survival analysis, including three patients who who died before the first disease response assessment.
þ Lines 172-176, Page 5, after revision
: The response to treatment was assessed in entire 34 patients, including three who died before the first response evaluation (two patients died of infection and one withdrawn from the study due to hypersensitivity to oxaliplatin). Five patients achieved PR and 16 patients obtained SD, while 13 patients did not respond to treatment. The ORR and DCR were 14.7% and 61.7%, respectively.
- Go S-I, Lee S-C, Bae WK, Zang DY, Lee HW, Jang JS, Ji JH, Kim JH, Park S, Sym SJ, et al. Modified FOLFIRINOX versus S-1 as second-line chemotherapy in patients with gemcitabine-failed metastatic pancreatic cancer: A randomized phase III trial (MPACA-3). Journal of Clinical Oncology. 2021;39(15_suppl):4119-.

Reviewer 4 Report
The author’s present a prospective study investigating the efficacy and safety of second line modified FOLFIRINOX in 34 patients with advanced biliary cancer, who had disease progression under gemcitabine-cisplatin treatment. The authors conclude that mFOLFIRINOX might offer a survival benefit in these patients, although the survival in the present study was slightly shorter than published literature. I think the paper is interesting and scientifically sound, I have a few comments on the paper, which are listed below:
Major:
- The authors report survival as the time from study inclusion. Perhaps this could explain the slightly shorter survival time compared with current literature. It could be considered to add a supplementary graph with survival from time of disease progression during gemcitabine-cisplatin treatment.
- The authors report three deaths due to infections within the study period. The authors state that the cause of death was infection in all three cases and thus not related to the study intervention. Did any of these patients have neutropenia due to mFOLFIRINOX? Couldn’t this be the primary cause that these patients acquired an infection in the first place (and thus relatable to the treatment)?
- Please specify types of dose reductions. Were only 25% dose reductions applied, or did some patients require 50-75% dose reductions? Was the 5-FU bolus emitted in other cases?
- The authors state that patients who received more cycles of FOLFIRINOX demonstrated longer PFS and OS times. This seems obvious, as mFOLFIRINOX was discontinued in the case of disease progression or death. Patients had to be alive to receive more cycles (i.e. immortal-time bias, Yadav et al. JAMA 2021). This statement could be strengthened by a multivariable analysis including number of cycles as a variable, or else should be removed from the manuscript.
- Were tumor markers (CEA, CA19-9) collected during follow-up? In addition, it could be interesting to add a waterfall plot to the manuscript to demonstrate tumor marker response in relation to objective disease response based on imaging.
Minor:
- Kaplan-Meier curves could benefit from numbers at risk.
- Minor spelling error in line 123 (mFOLFIRINOXSecondary).
Author Response
Reviewer #4:
The author’s present a prospective study investigating the efficacy and safety of second line modified FOLFIRINOX in 34 patients with advanced biliary cancer, who had disease progression under gemcitabine-cisplatin treatment. The authors conclude that mFOLFIRINOX might offer a survival benefit in these patients, although the survival in the present study was slightly shorter than published literature. I think the paper is interesting and scientifically sound, I have a few comments on the paper, which are listed below:
þ Thank you very much for your precious review of our manuscript. All the comments were very helpful for correction of this report, and we did our best to accommodate your detailed advice as much as possible. In accordance with the respectful comments, we have revised our manuscript.
Major:
1) The authors report survival as the time from study inclusion. Perhaps this could explain the slightly shorter survival time compared with current literature. It could be considered to add a supplementary graph with survival from time of disease progression during gemcitabine-cisplatin treatment.
þ We appreciate your comments on improvements to our study. You have raised an important point. Although we considered additional supplementary graph such as your comment, however to compare our survival outcomes with that of previous studies, we believe that it would be better to use the study enrollment time as a reference point of survival analysis as same. In addition, although the disease progressed after first-line GP therapy, there were six patients who took more than six months to start secondary treatment because there was no rapid deterioration. This may lead to confusion in survival analysis and interpretation of results.
2) The authors report three deaths due to infections within the study period. The authors state that the cause of death was infection in all three cases and thus not related to the study intervention. Did any of these patients have neutropenia due to mFOLFIRINOX? Couldn’t this be the primary cause that these patients acquired an infection in the first place (and thus relatable to the treatment)?
þ Thanks for your attentive comments. Of the three patients who died during study treatment, only one had grade 1 neutropenia, and the remaining two did not experience any grade of neutropenia. Thus, we considered that these three death events were less associated with chemotherapy itself.
3) Please specify types of dose reductions. Were only 25% dose reductions applied, or did some patients require 50-75% dose reductions? Was the 5-FU bolus emitted in other cases?
þ We had a total of 15 dose reduction cases. Of these, nine patients received one dose reduction (25% dose reduction) and the remaining six received only 50% of the total dose by reducing dose twice. In response, we provided additional information on dose reduction in the safety outcomes session. Furthermore, we skipped 5-FU bolus administration in all study populations according to study protocol.
þ Lines 192-194, Page 6, after revision
: In our study, 44% (n = 15) of patients received dose reductions, of which 40% (n = 6) experienced two dose reductions (50% reduction of total dose).
4) The authors state that patients who received more cycles of FOLFIRINOX demonstrated longer PFS and OS times. This seems obvious, as mFOLFIRINOX was discontinued in the case of disease progression or death. Patients had to be alive to receive more cycles (i.e. immortal-time bias, Yadav et al. JAMA 2021). This statement could be strengthened by a multivariable analysis including number of cycles as a variable, or else should be removed from the manuscript.
þ Thank you for your comments. In our study, patients who received less than three cycles of mFOLFIRINOX, so-called under-treated patients, mostly experienced rapid disease progression despite of treatment. However, patients with at least no disease progression with mFOLFIRINOX treatment achieved their disease control (at least SD/PR/CR) within a median of 1.3 months (95% CI, 1.1–1.5 months), and tended to maintain their disease status (median disease control 4.2 months (95% CI, 2.9–5.5 months). Thus, we believe that it is conceivable that they could be considered patients who benefited from mFOLFIRINOX treatment.
5) Were tumor markers (CEA, CA19-9) collected during follow-up? In addition, it could be interesting to add a waterfall plot to the manuscript to demonstrate tumor marker response in relation to objective disease response based on imaging.
þ We appreciate your comments on improvements to our study. We plotted the changes of tumor marker according to the best treatment response as a line graph with waterfall plot. Unfortunately, since we have not been able to derive a specific correlation, we only present it in the response letter
Left figure – best response and CEA Right figure – best response and CA 19-9
Minor:
1) Kaplan-Meier curves could benefit from numbers at risk.
þ According to your recommendation, we have provided the number at risk on survival curves. Please check the figures.
2) Minor spelling error in line 123 (mFOLFIRINOXSecondary).
þ Thank you for your detailed review. We corrected the spelling error.
þ Lines 132, Page 3, after revision
: mFOLFIRINOX. Secondary

Round 2
Reviewer 1 Report
Although some general concerns remain, the authors have satisfactorily answered most of the questions and concerns of the reviewer. I believe that this is an interesting and valid manuscript that has been strengthened by the additions and changes.
Reviewer 4 Report
The authors have improved their work. I have no further suggestions and recommend publication of the paper.